# Leaf litter decomposition rates increase with rising mean annual temperature in Hawaiian tropical montane wet forests

Lori D. Bothwell[1], Paul C. Selmants[2], Christian P. Giardina[3] and Creighton M. Litton[2]

[1] Natural Sciences Division, University of Hawaii at Hilo, Hilo, HI, USA
[2] Department of Natural Resources and Environmental Management, University of Hawaii at Manoa, Honolulu, HI, USA
[3] Institute of Pacific Islands Forestry, Pacific Southwest Research Station, USDA Forest Service, Hilo, HI, USA

## ABSTRACT

Decomposing litter in forest ecosystems supplies nutrients to plants, carbon to heterotrophic soil microorganisms and is a large source of $CO_2$ to the atmosphere. Despite its essential role in carbon and nutrient cycling, the temperature sensitivity of leaf litter decay in tropical forest ecosystems remains poorly resolved, especially in tropical montane wet forests where the warming trend may be amplified compared to tropical wet forests at lower elevations. We quantified leaf litter decomposition rates along a highly constrained 5.2 °C mean annual temperature (MAT) gradient in tropical montane wet forests on the Island of Hawaii. Dominant vegetation, substrate type and age, soil moisture, and disturbance history are all nearly constant across this gradient, allowing us to isolate the effect of rising MAT on leaf litter decomposition and nutrient release. Leaf litter decomposition rates were a positive linear function of MAT, causing the residence time of leaf litter on the forest floor to decline by ∼31 days for each 1 °C increase in MAT. Our estimate of the $Q_{10}$ temperature coefficient for leaf litter decomposition was 2.17, within the commonly reported range for heterotrophic organic matter decomposition (1.5–2.5) across a broad range of ecosystems. The percentage of leaf litter nitrogen (N) remaining after six months declined linearly with increasing MAT from ∼88% of initial N at the coolest site to ∼74% at the warmest site. The lack of net N immobilization during all three litter collection periods at all MAT plots indicates that N was not limiting to leaf litter decomposition, regardless of temperature. These results suggest that leaf litter decay in tropical montane wet forests may be more sensitive to rising MAT than in tropical lowland wet forests, and that increased rates of N release from decomposing litter could delay or prevent progressive N limitation to net primary productivity with climate warming.

Corresponding author
Paul C. Selmants,
selmants@hawaii.edu

## INTRODUCTION

Litter decomposition is a fundamental biogeochemical process influencing rates of carbon and nutrient cycling in forest ecosystems (*Perry, Oren & Hart, 2008*). Global syntheses indicate that temperature is a primary factor controlling litter decay rates (*Aerts, 1997*; *Gholz et al., 2000*; *Adair et al., 2008*), but these datasets are dominated by temperate forest ecosystems. The factors controlling litter decomposition in tropical wet forest ecosystems are less well studied (*Cusack et al., 2009*; *Wieder, Cleveland & Townsend, 2009*) and data from tropical montane wet forests are particularly sparse (*Malhi et al., 2010*). The paucity of litter decomposition data from tropical montane wet forests represents a significant gap in knowledge given recent evidence that litter decomposition in montane tropical wet forests may be more sensitive to rising temperature than in lowland tropical wet forests (*Waring, 2012*) and that warming in the tropics may be occurring faster at higher elevations (*Giambelluca, Diaz & Luke, 2008*).

One of the more tractable approaches to estimate the temperature sensitivity of litter decay in forests is through the use of elevation gradients, which can be powerful tools to examine climatic controls on ecosystem functioning (*Malhi et al., 2010*). Observational studies along elevation transects substitute space for time by examining litter decay rates in forests across a range of temperature environments, and so have the advantage of representing the long-term, integrated response of decomposition to changing mean annual temperature (MAT). Elevation gradients are seldom a perfect proxy for climate warming, however, because other factors that influence ecosystem processes may also vary with elevation, including plant species composition, precipitation and soil moisture, geologic substrate, and soil chemical and physical properties. These potentially confounding factors can complicate efforts to isolate the influence of temperature on ecosystem functioning along elevation gradients (*Wood, Cavaleri & Reed, 2012*).

We know of only two studies that have specifically examined the effect of temperature on leaf litter decomposition along elevation transects in the tropics (*Scowcroft, Turner & Vitousek, 2008*; *Salinas et al., 2011*), both of which indicate that temperature is a primary factor controlling leaf litter decomposition rates. However, the two studies vary widely in their estimates of the apparent $Q_{10}$ temperature coefficient, the proportional change in litter decay rate due to a 10 °C increase in temperature, potentially because of the confounding effects of precipitation and soil moisture, which also exert a strong control on litter decay (*Schuur, 2001*). For example, precipitation differences with elevation in *Salinas et al. (2011)* resulted in a three-fold variation in mean annual soil moisture across sites of different elevation. Relatively few litter decomposition studies and the wide range of results from those studies highlight the need for more research aimed at isolating the influence of temperature on leaf litter decay in the carbon dense and highly productive tropical montane wet forest biome.

Here we present results of a leaf litter decomposition experiment across a well-constrained 5.2 °C MAT gradient consisting of nine permanent plots in native-dominated tropical montane wet forests spanning 800 m elevation along the eastern slope of Mauna Kea Volcano on the Island of Hawaii. Many potentially confounding factors remain

constant across this MAT gradient, such as dominant canopy tree species, disturbance history, soil water content, geological substrate and soil type (*Litton et al., 2011*; *Iwashita, Litton & Giardina, 2013*; *Selmants et al., 2014*). We examined the decay of a common substrate (senescent *Metrosideros polymorpha* [Myrtaceae] leaves from a mid-elevation plot) across the MAT gradient over a six-month time period to address two main research questions: (i) Does rising MAT increase rates of leaf litter decomposition in tropical montane wet forests when other environmental factors are held constant?; and (ii) does variation in MAT alter the rate of nitrogen (N) release from decomposing leaf litter in tropical montane wet forests?

## MATERIALS AND METHODS

### Study sites

We tested the effect of rising MAT on leaf litter decomposition rates by using a highly constrained MAT gradient along the eastern flank of Mauna Kea volcano on the Island of Hawaii. This MAT gradient consists of nine permanent 20 × 20 m plots in native-dominated tropical montane wet forests spanning 800–1600 m in elevation, which corresponds to a 5.2 °C MAT gradient (13.8–18.2 °C). All nine MAT plots are similar in factors other than mean annual temperature that can affect ecosystem processes, including vegetation, disturbance history, soils, parent material, and soil water balance (*Litton et al., 2011*; *Iwashita, Litton & Giardina, 2013*; *Selmants et al., 2014*). Specifically, all nine MAT plots are in moderately aggrading mature forests with a canopy dominated by *M. polymorpha* and a mid-story dominated by three species of tree fern (*Cibotium* spp.). Soils are all well-drained Acrudoxic Hydrudands according to USDA Soil Classification System, and are all derived from ∼20,000 year-old volcanic ash deposits on top of a Pleistocene-age lava flow dominated by the minerals hawaiite and mugearite (*Wolfe & Morris, 1996*; *Litton et al., 2011*). Rainfall is not constant across the MAT gradient, but tends to decrease with elevation (Table 1). However, potential evapotranspiration also decreases with elevation (Table 1; *Giambelluca et al., 2014*), which results in near constant mean annual soil water content across the gradient (*Litton et al., 2011*). Solar radiation is also nearly constant across the MAT gradient (Table 1), and all MAT plots are below the trade wind inversion layer (*Cao et al., 2007*). The seven lower elevation MAT plots are in the Hawaii Experimental Tropical Forest and were accessed by permission from the USDA Forests Service and the State of Hawaii Department of Land and Natural Resources. The two highest elevation MAT plots are in the Hakalau Forest National Wildlife Refuge and were accessed with permission from the US Fish and Wildlife Service.

### Experimental design

We examined the effect of MAT on leaf litter decomposition by following the decay of senescent *M. polymorpha* leaves in each of the nine MAT plots for a six month period (June to December, 2012). Rainfall is fairly evenly distributed throughout the year on windward slopes in the Hawaiian Islands (*Giambelluca et al., 2013*) and seasonal changes in solar zenith angle, although much less pronounced than in higher latitude temperate

**Table 1** Environmental characteristics of the nine permanent plots along a 5.2 °C mean annual temperature gradient in tropical montane wet forests on the Island of Hawaii.

| MAT plot | Elevation (m) | Air temperature (°C)[a] | Rainfall (mm y$^{-1}$)[c] | Soil VWC: annual (m$^3$ m$^{-3}$)[a] | Soil VWC: experiment (m$^3$ m$^{-3}$)[b] | Potential evapo-transpiration (mm y$^{-1}$)[d] | Solar radiation (W m$^{-2}$ y$^{-1}$)[d] |
|---|---|---|---|---|---|---|---|
| 1 | 800 | 18.2 | 4,570 | 0.55 | 0.67 | 2,298 | 201.1 |
| 2 | 934 | 17.3 | 4,292 | 0.55 | 0.64 | 2,232 | 200.9 |
| 3 | 1,024 | 16.7 | 3,975 | 0.57 | 0.63 | 2,214 | 202.4 |
| 4 | 1,116 | 16.1 | 3,734 | 0.48 | 0.61 | 2,127 | 204.9 |
| 5 | 1,116 | 16.1 | 3,433 | 0.51 | 0.47 | 2,137 | 210.1 |
| 6 | 1,204 | 15.5 | 3,181 | 0.40 | 0.42 | 2,211 | 214.5 |
| 7 | 1,274 | 15.1 | 3,101 | 0.51 | 0.44 | 2,234 | 216.2 |
| 8 | 1,468 | 13.8 | 4,119 | 0.55 | 0.61 | 1,888 | 202.6 |
| 9 | 1,600 | 13.0 | 3,282 | 0.57 | 0.60 | 1,961 | 213.1 |

**Notes.**

[a] Mean annual air temperature and mean annual soil volumetric water content from *Litton et al. (2011)*.

[b] Mean monthly soil volumetric water content during the leaf litter decomposition experiment (June to December, 2012).

[c] Mean annual rainfall estimates from *Giambelluca et al. (2013)*.

[d] Mean annual potential evapotranspiration and solar radiation estimates from *Giambelluca et al. (2014)*.

regions, lead to slightly warmer air temperatures in summer months than in winter (*Giambelluca et al., 2014*).

We used intact senescent *M. polymorpha* leaves sorted from oven-dried litterfall samples collected monthly in elevated trays at 1116 m (16.1 °C; Table 1) from July 2010 through March 2012. These leaves were composited and well mixed for construction of individual litterbags. We used litter collected from one species at a single elevation as a standard substrate to isolate the effect of MAT on litter decay by minimizing variation in litter nutrient concentration. Each litterbag was 10 × 15 cm, constructed of fiberglass screen with 1.5 mm mesh size and contained ∼2.5 g of well-mixed senescent *M. polymorpha* leaves. We deployed 15 litterbags per MAT plot in June of 2012. We placed a cluster of three litterbags tethered with nylon fishing line to a central stake in five 5 × 5 m subplots within each MAT plot, resulting in five replicate litterbags for each collection period in each MAT plot. All litterbags were placed on the soil surface (top of the O horizon). We collected litterbags at one, three, and six months after initial placement. After each collection, litter was oven-dried at 70 °C, weighed and finely ground in a ball mill for chemical analysis. We did not correct for ash-free dry mass, but because they were placed on top of the undisturbed O horizon there was no evidence of soil accumulation on either litterbag surfaces or litter itself. Total N of initial, undecomposed leaf litter and of partially decomposed litter from the three collection periods was determined by combustion using a Costech Elemental Analyzer at the University of Hawaii at Hilo Analytical Laboratory (Costech Analytical Technologies; Valencia, CA USA).

## Data analysis

We estimated the decomposition rate of *M. polymorpha* leaf litter for each of the five replicate litterbag clusters within each MAT plot by fitting a single-pool negative

exponential model to the litter mass data (*Olson, 2007*) using non-linear regression with initial mass fixed at the measured value (*Adair, Hobbie & Hobbie, 2010*):

$$X(t) = e^{-kt} \tag{1}$$

where $X(t)$ is the proportional mass remaining at time $t$ (in days) and $k$ is the decomposition rate. The proportional initial mass, $X(0)$, is equal to 1 by definition and so is not estimated as a model parameter (*Adair, Hobbie & Hobbie, 2010*). The mean residence time of leaf litter on the forest floor was calculated as $1/k$ for each set of litterbags. We calculated the $Q_{10}$ temperature coefficient for litter decomposition rates across the MAT gradient as:

$$Q_{10} = (R_2/R_1)^{10/(T_2-T_1)} \tag{2}$$

where $Q_{10}$ is the proportional change in $k$ due to a 10 °C increase in MAT, $R_1$ and $R_2$ are regression-derived estimates of $k$ at the coolest and warmest MAT plots, and $T_1$ and $T_2$ are MAT values of the coolest and warmest MAT plots (13 °C and 18.2 °C, respectively). The proportion of initial N remaining at the end of the six-month incubation period was calculated by dividing the mass of N in litter collected after six months by the mass of N in the initial undecomposed litter (*Schuur, 2001*).

   We used ordinary least-squares linear regression to determine whether leaf litter decomposition rate ($k$), leaf litter residence time ($1/k$) and the percentage of leaf litter N remaining after six months varied significantly as a function of MAT. The plot ($n = 9$) was the smallest experimental unit to which the treatment (MAT) was applied, and so we use within-plot means of the five replicate litterbag clusters as the response variable for all linear regression analyses. We also used regression analysis at the plot level to determine whether annual rainfall and soil water content were significant predictors of leaf litter decay rates across the MAT gradient. We calculated 95% confidence intervals of the percentage of initial N remaining in leaf litter after one, three and six months to examine net N mineralization and net N immobilization at each stage of decomposition across the MAT gradient. For all statistical tests, we set $\alpha = 0.05$ and confirmed that the assumptions of normality and homoscedasticity were met. All statistical analyses were performed in R version 3.0.2 (*R Core Team, 2014*).

## RESULTS

Leaf litter decomposition rate ($k$) ranged from $1.67 \times 10^{-3} d^{-1}$ to $4.03 \times 10^{-3} d^{-1}$ across the MAT gradient and $k$ was a positive linear function of MAT ($R^2 = 0.65, p < 0.01$). Decomposition rates increased by $0.21 \times 10^{-3} d^{-1}$ for each 1 °C increase in MAT. Consequently, leaf litter residence time declined by 31 days for each 1 °C increase in MAT (Fig. 1). The estimated $Q_{10}$ for leaf litter decomposition was 2.17.

   Initial N concentration of mixed *M. polymorpha* litter was 8.5 mg g$^{-1}$. After six months of decomposition, the percentage of N remaining in decomposing *M. polymorpha* leaves declined significantly as a function of increasing MAT (Fig. 2), from ~88% of initial N at the coolest site to ~74% of initial N at the warmest site, a decline of approximately two percentage points for each 1 °C increase in MAT. Nitrogen remaining in leaf litter was never

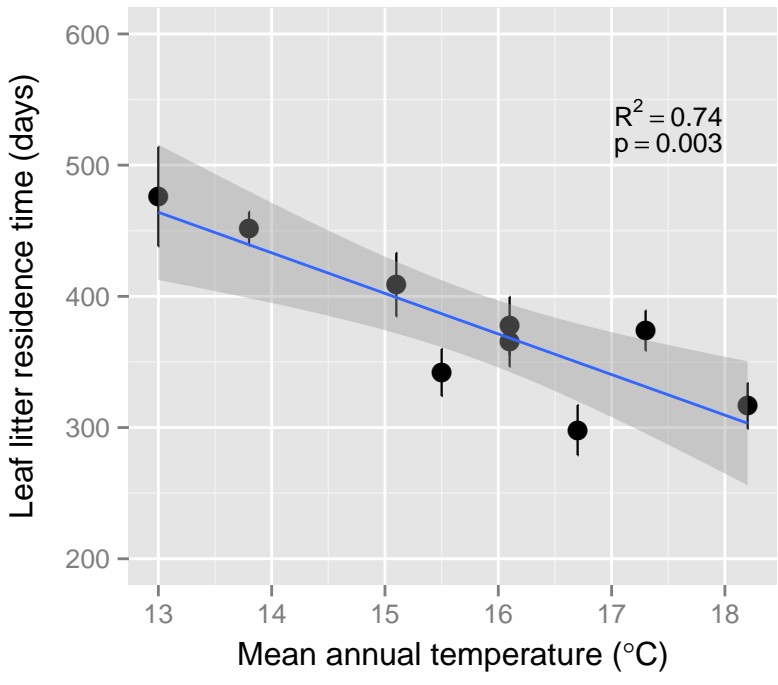

**Figure 1 Leaf litter residence time across a mean annual temperature gradient on the Island of Hawaii.** Residence time ($1/k$) of *Metrosideros polymorpha* leaf litter from a common site across a 5.2 °C mean annual temperature gradient in Hawaiian tropical montane wet forests. Black circles are means and error bars represent $\pm 1$ S.E.; *n* = 5 replicates per MAT plot. The blue line represents the linear best fit (residence time = $-30.92$*MAT + 866.05) and the gray shaded area represents the 95% confidence interval around the best-fit line.

significantly larger than 100% at any stage of decomposition within any of the nine MAT plots (Fig. 3), indicating there was no net N immobilization in decaying leaf litter. Neither annual rainfall nor soil water content during the six month experiment period (Table 1) were significant predictors of leaf litter decay rates ($R^2 = 0.013$, $p = 0.34$ for annual rainfall; $R^2 = 0.03$, $p = 0.67$ for soil water content).

## DISCUSSION

Quantifying the temperature sensitivity of leaf litter decomposition and nutrient release is critical to understanding how forest ecosystem processes will respond to climate change. We used a highly constrained MAT gradient to demonstrate that both mass loss and N release during leaf litter decay increase linearly in response to rising MAT in Hawaiian tropical montane wet forests. Globally, there is evidence that leaf litter decay is slowed by cool temperatures in tropical montane forests (*Waring, 2012*) and that climate warming in the tropics is occurring faster at higher elevations (*Bradley, Keimig & Diaz, 2004*; *Giambelluca, Diaz & Luke, 2008*). When combined with results presented here, this evidence suggests leaf litter decomposition rates in tropical montane wet forests may increase substantially with climate warming in the coming decades.

The rates of leaf litter decomposition reported here ($1.67 \times 10^{-3} d^{-1}$ to $4.03 \times 10^{-3} d^{-1}$ across the MAT gradient) are consistent with other studies in montane wet forests in

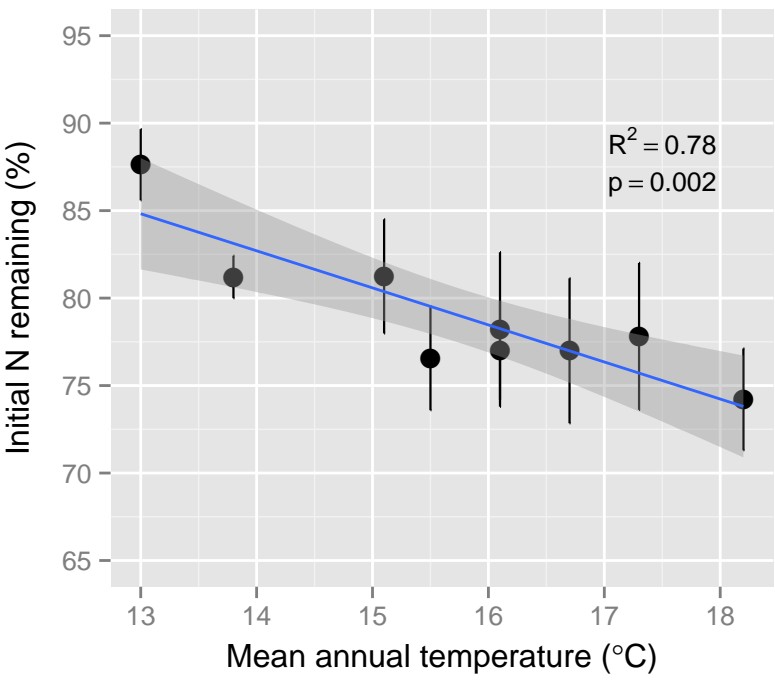

**Figure 2 Nitrogen remaining in leaf litter after six months of decomposition across a mean annual temperature gradient on the Island of Hawaii.** The percentage of initial nitrogen (N) remaining in *Metrosideros polymorpha* leaf litter from a common site after six months of decomposition across a 5.2 °C mean annual temperature gradient in Hawaiian tropical montane wet forests. Black circles are means and error bars represent ±1 SE; $n = 5$ replicates per MAT plot. The blue line represents the linear best fit (N remaining $= -2.12*$MAT $+ 112.36$) and the gray shaded area represents the 95% confidence interval around the best-fit line.

Hawaii (*Hobbie & Vitousek, 2000*; *Schuur, 2001*; *Scowcroft, Turner & Vitousek, 2008*) and well within the two orders of magnitude range of $3 \times 10^{-4} d^{-1}$ to $3 \times 10^{-2} d^{-1}$ reported for tropical wet forests globally (*Gholz et al., 2000*; *Cusack et al., 2009*; *Waring, 2012*). There are a number of factors aside from temperature that can affect leaf litter decomposition rates in tropical montane wet forests, most notably leaching from precipitation (*Wieder, Cleveland & Townsend, 2009*), soil oxygen availability related to soil water content (*Schuur, 2001*) and leaf litter chemistry (*Wieder, Cleveland & Townsend, 2009*; *Salinas et al., 2011*). By decomposing a common substrate across a highly constrained MAT gradient, we were able to isolate the effect of temperature by largely controlling for the effects of precipitation, soil moisture, initial litter chemistry and other potential confounding factors, demonstrating a strong linear increase in leaf litter decomposition rate with rising MAT. This increase in leaf litter decay rate is in line with, and likely contributes to, the substantial increase in soil-surface $CO_2$ efflux across this MAT gradient (*Litton et al., 2011*). Notably, soil organic carbon storage remains constant across the MAT gradient despite increased rates of litter decay and soil-surface $CO_2$ efflux with rising MAT (*Selmants et al., 2014*; *Giardina et al., 2014*).

The $Q_{10}$ temperature coefficient, which describes the rate of change in a biological or chemical process over a 10 °C interval, often falls within the range of 1.5–2.5 when applied

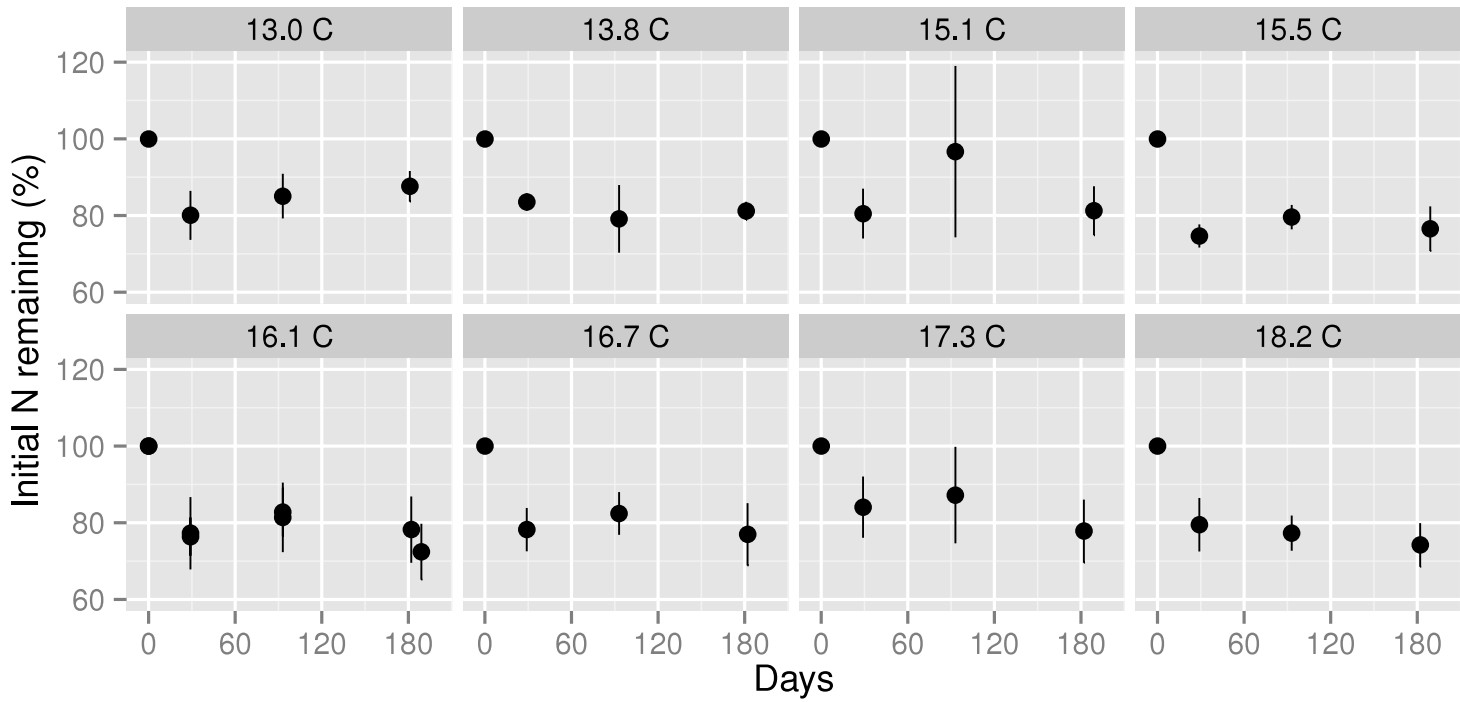

**Figure 3 Nitrogen remaining in decomposing leaf litter at three stages of decomposition across a mean annual temperature gradient on the Island of Hawaii.** Nitrogen remaining in *Metrosideros polymorpha* leaf litter at each stage of decomposition across a 5.2 °C mean annual temperature gradient in Hawaiian tropical montane wet forests. Black circles are means with error bars representing the 95% confidence interval around the mean; $n = 5$ litterbags per MAT plot. Values below 100% indicate net N mineralization from decomposing leaf litter; values above 100% indicate net N immobilization in decomposing leaf litter. The header of each panel indicates the mean annual temperature (in °C) of each plot. Note that data from the two 16.1 °C plots are plotted in the same panel.

to heterotrophic processes such as organic matter decomposition (*Reiners, 1968*; *Kätterer et al., 1998*; *Gholz et al., 2000*; *Hyvönen, Ågren & Dalias, 2005*; *Zhou et al., 2009*). Our $Q_{10}$ estimate for leaf litter decomposition (2.17) is within this range, and is similar to the $Q_{10}$ estimate for soil-surface $CO_2$ efflux (2.26) across the same MAT gradient (*Litton et al., 2011*), suggesting a consistent response of carbon cycling rates to rising temperature in tropical montane wet forests. In contrast to our results, *Salinas et al. (2011)* estimated a $Q_{10}$ for leaf litter decay of 3.06 in five tropical forest plots across an elevation gradient spanning 12.8 °C (11.1–23.9 °C). Although there is wide variation in estimates of apparent $Q_{10}$ from litter decomposition studies (*Gholz et al., 2000*; *Adair et al., 2008*), we suggest the *Salinas et al. (2011)* estimate of litter decay $Q_{10}$ is ∼40% higher than ours largely because of confounding factors related to site selection and data analysis technique. First, the lowest temperature site in the *Salinas et al. (2011)* study was also the driest, with a mean annual soil moisture nearly three-fold lower than the mean of the other four sites, potentially depressing $k$ for this low temperature site. In contrast, mean annual soil moisture is nearly constant across our MAT gradient (*Litton et al., 2011*). Although there was some variation in soil water content among our MAT plots during the six-month period when leaf litter was decomposing, the driest plots during the experiment period were in the middle of our gradient and the percentage variation in soil moisture was an order of magnitude less than that of

*Salinas et al. (2011)*. Second, *Salinas et al. (2011)* used linear regression of log-transformed mass loss data to estimate decomposition rates, which can result in sizable overestimates of $k$ depending on error structure (*Adair, Hobbie & Hobbie, 2010*). We used non-linear regression of untransformed decomposition data, which consistently yields accurate $k$ estimates (*Adair, Hobbie & Hobbie, 2010*). The potential confounding effect of a three-fold variation in soil moisture combined with the use of log-transformed mass loss data in the *Salinas et al. (2011)* study suggest that their estimate of apparent $Q_{10}$ may be artificially inflated.

We found no evidence of N limitation to leaf litter decomposition across the MAT gradient. At all stages of decomposition, the proportion of initial N remaining in decomposing leaf litter never significantly exceeded 100% at any of the MAT plots. This evidence is consistent with results from a fertilization experiment at a site near the middle of our MAT gradient, in which individual additions of N and phosphorus (P) had no effect on *M. polymorpha* leaf litter decomposition and combined N and P additions had only a weak positive effect (*Hobbie & Vitousek, 2000*). We also found an overall trend of increased rates of N release from decomposing litter with rising MAT. Taken together, this evidence suggests that warming will increase rates of N cycling and availability in these forests, a response consistent with results from warming experiments across a wide range of forest and grassland ecosystems (*Rustad et al., 2001*). We did not measure leaf litter phosphorus (P) dynamics in this study, so it remains unclear how rates of P release from decomposing leaf litter respond to rising MAT. However, since P does not directly limit *M. polymorpha* leaf litter decomposition at a site near the middle of our gradient (*Hobbie & Vitousek, 2000*), we consider it likely that increasing rates of leaf litter decay with rising MAT will increase rates of P release and availability similar to the trend for N documented here.

Results from this leaf litter decomposition experiment across a well-constrained MAT gradient have two potentially countervailing implications for how the carbon balance of tropical montane wet forests will respond to climate warming, at least within the MAT range studied here and where increasing temperature does not drive significant, concomitant changes in leaf litter chemistry or soil water balance. First, our results indicate that warming will increase rates of leaf litter decay in tropical montane wet forests, which explains part of the warming-induced increase in rates of soil-surface $CO_2$ efflux to the atmosphere (*Litton et al., 2011*). At the same time, these results suggest that more rapid decomposition with warming should also increase rates of nutrient release from decaying leaf litter, at least in forest ecosystems where decomposition is not currently nutrient limited. The availability of nutrients may strongly regulate whether ecosystem carbon sequestration keeps pace with rising atmospheric $CO_2$ concentrations (*Luo et al., 2004*). An increase in rates of nutrient release from decaying leaf litter with climate warming, as suggested by our results, could delay or even prevent the onset of progressive nutrient limitation of ecosystem carbon sequestration.

## ACKNOWLEDGEMENT

Thanks to Kupu Hawaii Youth Conservation Corps interns Nathanael Friday and Isaac Ito for assistance with data collection.

### Funding

This study was funded by the National Science Foundation through the Pacific Internship Program for Exploring Science (PIPES; BIO-1005186) and the Ecosystem Science Program (DEB-0816486). Additional funding was provided by the USDA Forest Service, Institute of Pacific Islands Forestry, Pacific Southwest Research Station (Research Joint Ventures 09-JV-11272177-029 and 12-JV-11272139-047) and the College of Tropical Agriculture and Human Resources, the University of Hawaii at Manoa via the USDA National Institute of Food and Agriculture, Hatch and McIntire-Stennis Programs (HAW00132-H, HAW01127-H, HAW00188-M, and HAW01123-M). The funders had no role in study design, data collection and analysis, decision to publish, or preparation of the manuscript.

### Grant Disclosures

The following grant information was disclosed by the authors:
Pacific Internship Program for Exploring Science (PIPES): NSF BIO-1005186.
National Science Foundation, Ecosystem Science Program: NSF DEB-0816486.
USDA Forest Service, Institute of Pacific Islands Forestry, Pacific Southwest Research Station Research Joint Ventures: 09-JV-11272177-029, 12-JV-11272139-047.
College of Tropical Agriculture and Human Resources, University of Hawaii at Manoa via the USDA National Institute of Food and Agriculture, Hatch and McIntire-Stennis Programs: HAW00132-H, HAW01127-H, HAW00188-M, HAW01123-M.

### Competing Interests

Christian P. Giardina is an employee of the USDA Forest Service.

### Author Contributions

- Lori D. Bothwell and Paul C. Selmants conceived and designed the experiments, performed the experiments, analyzed the data, wrote the paper, prepared figures and/or tables, reviewed drafts of the paper.
- Christian P. Giardina and Creighton M. Litton conceived and designed the experiments, contributed reagents/materials/analysis tools, reviewed drafts of the paper.

### Field Study Permissions

The following information was supplied relating to field study approvals (i.e., approving body and any reference numbers):

Permission to access the Hawaii Experimental Tropical Forest was provided by the USDA Forest Service and the State of Hawaii Department of Land and Natural Resources Division of Forestry and Wildlife through approval of an annual permit application. Permission to access the Hakalau Forest National Wildlife Refuge was provided by the US Fish and Wildlife Service through approval of an annual permit application. Neither of these permits are individually numbered.

## Supplemental Information

Supplemental information for this article can be found online at http://dx.doi.org/10.7717/peerj.685#supplemental-information.

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
