# Peer review of "Leaf litter decomposition rates increase with rising mean annual temperature in Hawaiian tropical montane wet forests"

_PeerJ, doi:10.7717/peerj.685_

## Round 0.1 · original submission · Minor Revisions

· Academic Editor

Minor Revisions

I have read the comments from the three reviewers and also the ms by myself, and I would agree with the reviewers that minor revisions be made before a firm decision can be made. In addtion to those comments made by the reviewers, the authors should also regress the leaf litter residence time (Figure 1), initial N remaining (Figure 2) and percentage of initial N remaining (Figure 3) against the mean annual temperature (Figures 1 and 2) or decomposition days (Figure 3), with individual measurement data, not using the only means (plus SEs), since using the means only would artificially increase the R or R square with much less variations among the individual site data. Futuremore, the authors would need to address those comments of the reviewers carefully.

Reviewer 1 ·

Basic reporting

This study examines how temperature alters litter decomposition rate using an elevation gradient. It found that both litter decay and N loss increased with temperature. The manuscript is very well written and easy to read.

Experimental design

Research questions are clearly defined and experiment design is sound. I can tell that the authors were aiming for brevity; however, additional details can be added to clarify some confusing points, including
L113, the litter were degraded from June to December (?). Why was the study conducted during this period? How do environmental factors vary during this period compared to the other half of the year?
L114, were litter bags placed on the soil surface?
L117, were the mass loss data controlled for ash content?
L124 states that litter data were pooled to calculate one k value at each site. k in Fig.1, however, had estimates of SE. How was k replicated?

Validity of the findings

Again, the authors were targeting brevity in results and discussion. I would argue that some additional data and discussion should be added to support their conclusions. Even though raw data of mass loss were included in the supplementary materials, I had a hard time figuring out how well the exponential model fits the mass loss data. It would be useful to have information of the model fitting in the results. It would also be useful to have the mass loss data in a more processed form in supplementary materials.

The authors did not discuss much about other confounding factors, except for saying that they controlled for them (L177-179). I am afraid that they might miss an opportunity to strengthen their study. For example, I guess there was a negative relationship between litter residence time and MAP, given the data in Table 1 and Figure 1. However, Schuur et al. (2001) found that high rainfall limited litter decay through lowering soil oxygen availability in montane rainforests in Hawaii. What can these seemingly contradicting results tell us? Does it mean that this study has a good control for soil water status? Again, instead of shying away from the confounding factors, the authors should fully take advantage of their dataset.

Following the above comment, I am curious about how well the experiment controlled for soil water content (SWC). The SWCs in Table 1 are annual averages. The experiment, however, only lasted 6 months between June to December. How consistent were SWC among sites during this period?

Comments for the author

L32, “all stages of decomposition” sounds as if this was a long-term study.
L236-238, this statement is a bit too strong. The N pattern in Fig 2 seems to be driven mostly by the point around MAT of 13 degree C. If this point is removed, the pattern is not very strong. We also don’t know in which forms N was lost during litter decay and whether it was available for plant uptake.

Reviewer 2 ·

Basic reporting

This article is well structured and written. The objectives are very clear with sufficient introduction and background. It helps to understand the response of litter decomposition to single factor of rising temperature in tropical forests.

Experimental design

The experiment is generally explicitly designed. Research has been appropriately conducted.

Validity of the findings

The data analyses are sound and results are clear.

·

Basic reporting

No comments

Experimental design

This manuscript deals with the effect of temperature on litter decomposition in a wet montane tropical forest, measured through an elevation gradient. The manuscript is concise and well-written and makes a valid contribution to the field of decomposition studies.

Validity of the findings

Although I acknowledge the fact that no proxy for climate change is perfect and that the altitudinal gradient used in this study is highly constrained, I would have liked to see more discussion about the potential effects of precipitation and microbial communities on the decomposition process. The leaching effect of rainfall and the effect of microbial biomass, even when not measured, should be at least incorporated to the discussion in the context of climate change. Was precipitation measured? Could it be included in the regression to identify its contribution?

Why were leaves collected from one plot only and not from each plot if the plant was chosen due to its presence along the whole gradient?

Some of the arguments in the Discussion section do not seem to be logically connected. For example, p. 8, l. 165-169: the authors state that a previous study reported a weak relationship between decomposition rates and mean annual temperature (MAT) in tropical wet forests but mention that this report is based on data from montane forests with MAT < 20°C. First of all, the data obtained in Waring (2012) come from tropical wet forests but not necessarily montane forests. The authors then conclude that their results combined with the results from Waring’s study indicate that litter decomposition is strongly linked to temperature. I do not follow the logic here. Could you please clarify your argumentation?

Discussion p.10: what would be the consequences at the ecosystem level of increased N availability, knowing that the system is not N-limited? Increased rates of N release from decomposition do not necessarily imply increased rates of N cycling (l. 216), especially when there is no N limitation in the system.

Comments for the author

Specific comments:
- P.2, l.20: “more warming faster”?
- P.4, l. 80-81: please add the family of Metrosideros polymorpha as not all lectors will be familiar with this plant. Also, there is a typo in the spelling of the species (on p.6, l.107 as well).
- P.5, l. 97-98: what is the soil classification that the authors are using?
- P.7, l.143: “homoscedasticity”, not “homoskedasticity”
- P.9, l.176: Schuur (2001) does not report that soil moisture is a main driver of decomposition but rather than precipitation and soil oxygen availability are important factors. This is why I think the authors should discuss the influence of precipitation on litter decomposition, even when soil moisture is nearly constant throughout the gradient.
- P9, l.193: Salinas et al. (2011) and not (2010)

---

## Round 0.2 · accepted · Accept

· Academic Editor

Accept

The revised ms is now acceptable for publication.